# Influence of Precipitation Hardening on the Mechanical Properties of Co-Cr-Mo and Co-Cr-W-Mo Dental Alloys

**Marko Sedlaček** [1] , **Katja Zupančič** [2] , **Barbara Šetina Batič** [1] , **Borut Kosec** [2] , **Matija Zorc** [2] **and Aleš Nagode** [2,*]

[1] Institute of Metals and Technology, Lepi Pot 11, 1000 Ljubljana, Slovenia
[2] Department for Materials and Metallurgy, Faculty of Natural Sciences and Engineering, University of Ljubljana, Aškerčeva 12, 1000 Ljubljana, Slovenia
* Correspondence: ales.nagode@ntf.uni-lj.si; Tel.: +386-1-20-00-433

**Abstract:** Co-Cr alloys have good mechanical properties such as high hardness, excellent magnetic properties and good corrosion resistance. For this reason, they are most commonly used as dental and orthopaedic implants. Generally, cast Co-Cr-Mo alloys and forged Co-Ni-Cr-Mo alloys are used for the production of implants. In this study, we investigated two dental alloys, namely, Co-Cr-Mo and Co-Cr-W-Mo alloys. The aim of this study was to determine the effect of heat treatment on the development of the microstructure and to evaluate its influence on the alloys' mechanical and tribological properties. The samples were first solution-annealed at 1200 °C in an argon atmosphere for 2 h, then quenched in water and subsequently aged at 900 °C in an argon atmosphere for 1, 3 and 12 h. A microstructural analysis was performed using SEM, with EDS for microchemical analysis and EBSD for phase identification. In addition, the Vickers hardness and wear resistance of the two alloys were analysed before and after heat treatment. The Co-Cr-Mo alloy showed better wear resistance and also a generally higher hardness than the Co-Cr-W-Mo alloy. Both alloys showed signs of abrasive and adhesive wear, with carbide particles detaching from the Co-Cr-W-Mo alloy due to the lower hardness of the matrix. The Co-Cr-Mo alloy showed the best abrasion resistance after the longest aging time (12 h), while the Co-Cr-W-Mo alloy showed the best as-cast abrasion resistance. With ageing, the wear resistance of both alloys increased.

**Keywords:** Co-Cr alloys; microstructure analysis; hardness; wear resistance



## 1. Introduction

Co-Cr-Mo and Co-Cr-W-Mo alloys are commonly used in biomedical applications such as joint replacement prosthetics and dental implants due to their excellent corrosion resistance, wear resistance and biocompatibility.

Co-Cr-Mo alloys are typically composed of about 60–70% cobalt, 25–30% chromium and 5–7% molybdenum, while Co-Cr-W-Mo alloys are typically composed of about 50–65% cobalt, 25–35% chromium, 5–20% tungsten and 0–10% molybdenum, although their exact composition may vary depending on the specific application [1,2].

In general, Co-Cr-Mo alloys have two crystal structures, hexagonal close-packed (ε-phase) and face-centred cubic (γ-phase). However, the transformation of the γ-phase to the ε-phase is extremely slow under normal cooling conditions, so that the high-temperature γ-phase structure is normally present at room temperature. This is also the case for pure cobalt, which shows an allotropic transformation to a γ-phase crystal structure at 400 °C [3]. Chromium, molybdenum [4] and tungsten [3] increase the transformation temperature, as both increase the stability of the ε-phase. Anyway, with isothermal aging in the temperature region between 650 and 950 °C, the transformation from γ-phase to ε-phase can be completed [5].

In the as-cast state, the microstructure of the Co-Cr-Mo alloy usually consists of a dendritic, metastable γ-FCC matrix due to the sluggish nature of the γ-phase → ε-phase

transformation. The microstructure also consists of secondary phases, mainly blocky carbides of the $M_{23}C_6$ type, which mostly precipitate at the grain boundaries and in the interdendritic regions. In general, $M_{23}C_6$ carbides have been identified as the most important secondary phase in Co-Cr alloys [4]. The mechanical properties can be improved by heat treatment through precipitation hardening, i.e., dissolving the large carbide network and forming fine precipitates during aging.

The wear resistance of materials used in medical applications is important because abrasion particles can have biological or toxicological effects if swallowed or inhaled. Therefore, the strength, hardness and wear resistance of alloys are also of great importance. The strength and resistance of cobalt-based alloys to abrasive wear depends primarily on the volume fraction of carbides and their morphology. The morphology of carbides depends mainly on the alloy and on the formation and size of the formed carbides, i.e., mainly on the solidification conditions of the alloy. Large hypereutectic carbides are good for abrasive wear resistance but bad for the overall strength of the alloy. In general, the smaller the carbides, the better the abrasive wear resistance. Chiba [6] reported that abrasive wear is induced by precipitates such as $\gamma$-phase precipitates. He also reported that Co-Cr-Mo alloys containing the $\varepsilon$-phase (formed by martensitic transformation) have a significantly better wear resistance.

By adding different alloying elements into Co-Cr alloys, the mechanical properties can be altered [7]. One of these elements is tungsten, which has a strong tendency to form carbides and other precipitates that can give rise to the formation of the $\varepsilon$-phase during cooling. Tungsten also increases the transformation temperature and the stability of the $\varepsilon$-phase [8]. Adding tungsten to Co-Cr-Mo alloys increases the tensile and yield strength, as well as the hardness, of the alloy [9]. It can have also beneficial influence on corrosion and wear resistance [9,10]. Adding tungsten to the Co-Cr-Mo alloy composition promotes the formation of long straight lines of $\varepsilon$-phase in the Co-Cr-W-Mo alloy.

Beside the chemical composition, the mechanical properties of Co-Cr-based alloys can be significantly improved by heat treatment [4,11]. The most typical heat treatment is precipitation hardening, which takes place in three steps: solution annealing, quenching and aging. Precipitation or age hardening is a heat treatment that results in the formation of evenly distributed fine precipitates in the metal matrix. The aim of precipitation hardening is to ensure and improve the strength and hardness of the alloy. Fine precipitates in the microstructure impede dislocation movements by forcing the dislocations to either cut through or go around the precipitated particles. By restricting the dislocation movements during deformation, the alloy is strengthened. However, the strengthening effect is mainly influenced by the interdistance and size of the precipitates. This can be determined by the ageing conditions, i.e., the ageing temperature and, especially, the ageing time. In general, longer aging times result in higher strength and hardness due to the increased precipitation of the $\gamma'$ phase to reach the optimum size. However, this also results in a decrease in ductility and toughness. After that the alloy is overaged: the precipitates are coarsened and the strengthening effect is reduced, while ductility is increased. In the case of Co-Cr-W-Mo alloys, the long straight lines of $\varepsilon$-phase can change in morphology, size and distribution during aging [12]. The aging process can lead to the coarsening or dissolution of the $\varepsilon$-phase, which can affect the alloy's mechanical properties.

In this investigation, the effect of precipitation hardening on microstructure development, hardness and wear properties of two commercially available dental alloys (Co-Cr-Mo and Co-Cr-W-Mo) was investigated. The alloys were investigated at three different times of aging and compared to the alloys in as-cast and quenched conditions.

## 2. Materials and Methods

### 2.1. Material

In this study, two commercially available alloys, Co-Cr-Mo and Co-Cr-W-Mo, were investigated. The chemical composition of the investigated materials in wt.% is shown in Table 1. The chemical composition was determined with the X-ray fluorescence spec-

trometer XRF (Thermo Scientific Niton XL3t GOLDD+, Thermo Fisher Scientific, Waltham, MA, USA). It can be seen that both materials contained similar proportions of Co and Cr, with Co-Cr-W-Mo also contained W and Mo. The alloys were in as-cast round bars with a diameter of 6 mm and a height of 17 mm.

**Table 1.** Chemical composition of the investigated materials in wt.%.

|  | Co | Cr | Mo | Mn | Fe | Si | W | Nb | C |
|---|---|---|---|---|---|---|---|---|---|
| Co-Cr-Mo | 66.0 | 28.4 | 6.2 | 0.4 | 0.3 | 0.64 | - |  | <1 |
| Co-Cr-W-Mo | 63.5 | 24.2 | 2.7 | 0.1 |  |  | 8.8 | 0.36 |  |

*2.2. Heat Treatment*

Both alloys were subjected to precipitation hardening consisting of solution annealing, quenching and artificial ageing. Solution annealing was carried out in a GSL 1700X tube furnace in an argon protective atmosphere at a temperature of 1200 °C for 2 h. The material was then quenched in water and aged. Aging was conducted under an argon protective atmosphere in the same furnace as that of solution annealing. The aging temperature was set at 900 °C, and the holding time varied between 1 h, 3 h and 12 h. Figure 1 shows a schematic representation of the heat treatment performed. For comparison, the as cast and solution annealing states were also analysed.

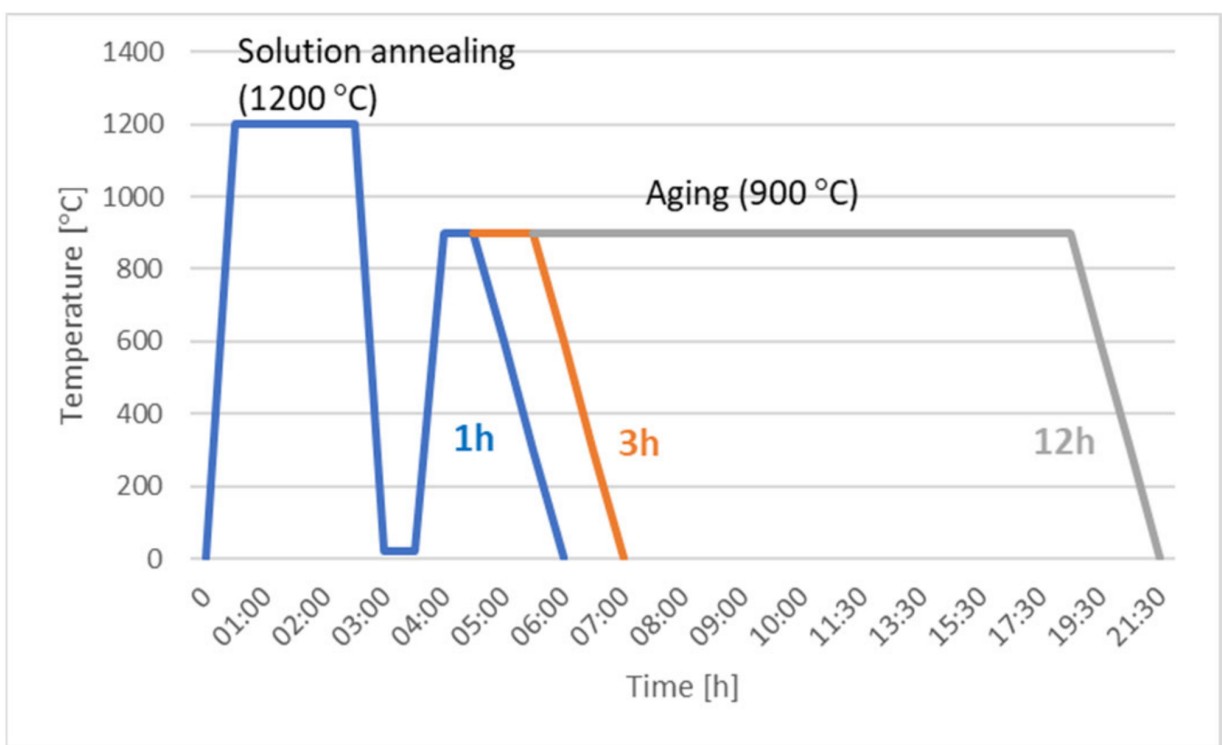

**Figure 1.** Schematic representation of the performed heat treatment.

*2.3. Hardness Measurements*

The hardness of the alloys studied was measured by the Vickers method according to the standard ISO 6507-1 using the Instruon Tukon 2100B instrument, Instron, Norwood, MA, USA. The hardness was measured on metallographically prepared samples in the centre of the rolls. All hardness measurements were carried out with load HV1.

### 2.4. Microstructure Analysis

For each sample, metallographic specimens were prepared by grinding and polishing. For grinding, different types of abrasive paper with granulation of 320, 500, 800, 1200 and 4000 were used for 5 min each. Polishing was performed with two different fabrics, i.e., Mol with a 3 μm diamond suspension and Nap with a 1 μm diamond suspension. The microstructures were analysed under a scanning electron microscope (SEM) (FEG-SEM Thermo Scientific Quattro S, Thermo Fisher Scientific, Hillsboro, OR, USA). Microchemical analyses were performed using energy X-ray spectroscopy (EDS) with an Ultim Max 65 mm$^2$ detector (Oxford Instruments, Abingdon, UK). Secondary and backscattered electrons were used for imaging. An electron beam accelerating voltage of 20 kV was used for imaging, and energy-dispersive X-ray spectroscopy (EDS). EBSD analyses (JEOL JSM6500F with OXFORD HKL Channel 5, Tokyo, Japan) were conducted for phase identification.

### 2.5. Wear Measurements

The wear resistance of the alloys studied was measured on a Hitman reciprocating machine with a ball on disc configuration. The tests were carried out under dry sliding conditions and a contact pressure of 0.8 GPa (30 N). An $Al_2O_3$ ball with a diameter of 20 mm was used as a counterpart in order to concentrate all the wear on the material under investigation. During sliding, the coefficient of friction was recorded. The tests were limited to a sliding time of 15 min, which corresponded to a sliding distance of 7.2 m. After the test, the volumes of the wear tracks were measured with a 3D optical profilometer (Alicona Infinitefocus G4, Raaba-Grambach, Austria). To ensure the repeatability of the tests, at least three replicates were analysed for each sample. The specific wear rate was calculated for all wear measurements. The specific wear rate is defined as the wear volume divided by the product of the normal load and the sliding distance.

## 3. Results and Discussion

### 3.1. Microstructure Characterisation

#### 3.1.1. Co-Cr-Mo Alloy

Figure 2, which shows the Co-Cr-Mo in the as-cast state, reveals a typical microstructure with dendritic morphology and crystal segregation. The EDS analysis confirmed that the matrix (γ-phase) was a solid solution alloy rich in cobalt, chromium and molybdenum. Dendrites were also present in the microstructure and were mainly located in the interdendritic region and at the crystal boundaries. The EDS analysis confirmed the increased content of chromium, molybdenum and carbon. The EBSD analysis (Figure 3) confirmed the presence of $M_{23}C_6$ carbides, where M stands for Cr. This is also consistent with literature sources [13,14]. Individual dark particles were described as inclusions by Giacchi [3]. With the EDS analysis, it was determined that these were $SiO_2$ inclusions, as an increased proportion of silicon and oxygen was detected in their composition.

As can be seen in Figure 2b, solution annealing and quenching in water led to a more homogeneous microstructure without crystal segregations. The number of carbides decreased, but their composition was slightly changed. The EDS analysis of the carbide area showed that the cobalt and molybdenum content decreased, while the chromium content increased significantly, by about 20 wt.%. Nevertheless, from the results of the EDS analyses and the literature, we could conclude that these were $M_{23}C_6$ carbides.

As can be seen in Figure 2c, after an ageing of 1 h, no significant changes in the microstructure could be detected compared to the quenched state. After 3 h of ageing, small precipitates formed during ageing could be seen in the microstructure (Figure 2d). As can be seen, carbides precipitated within the crystal grains and along the grain boundaries. According to the EBSD analysis (Figure 3), these were $M_{23}C_6$ carbides, which is consistent with a literature review [3]. The contents of chromium and cobalt in the FCC-α matrix continued to decrease slightly, by about 3 wt.% for both elements.

Aging for 12 h resulted in the formation of coarser and much rounder precipitates, but their number decreased (Figure 2e), which can be attributed to Ostwald ripening. As can

be seen in Figure 4, the EBSD analysis confirmed the presence of $Cr_{23}C_6$ carbides. Lighter molybdenum-based carbides—$M_6C$ carbides—could also be observed at the boundaries of the crystal grains, which was also noted by Taylor and Waterhouse [15]. Of the elements in the alloy, molybdenum has the highest Z number (Z contrast), which is why its microstructure has a lighter colour. On the other hand, chromium carbides appear darker.

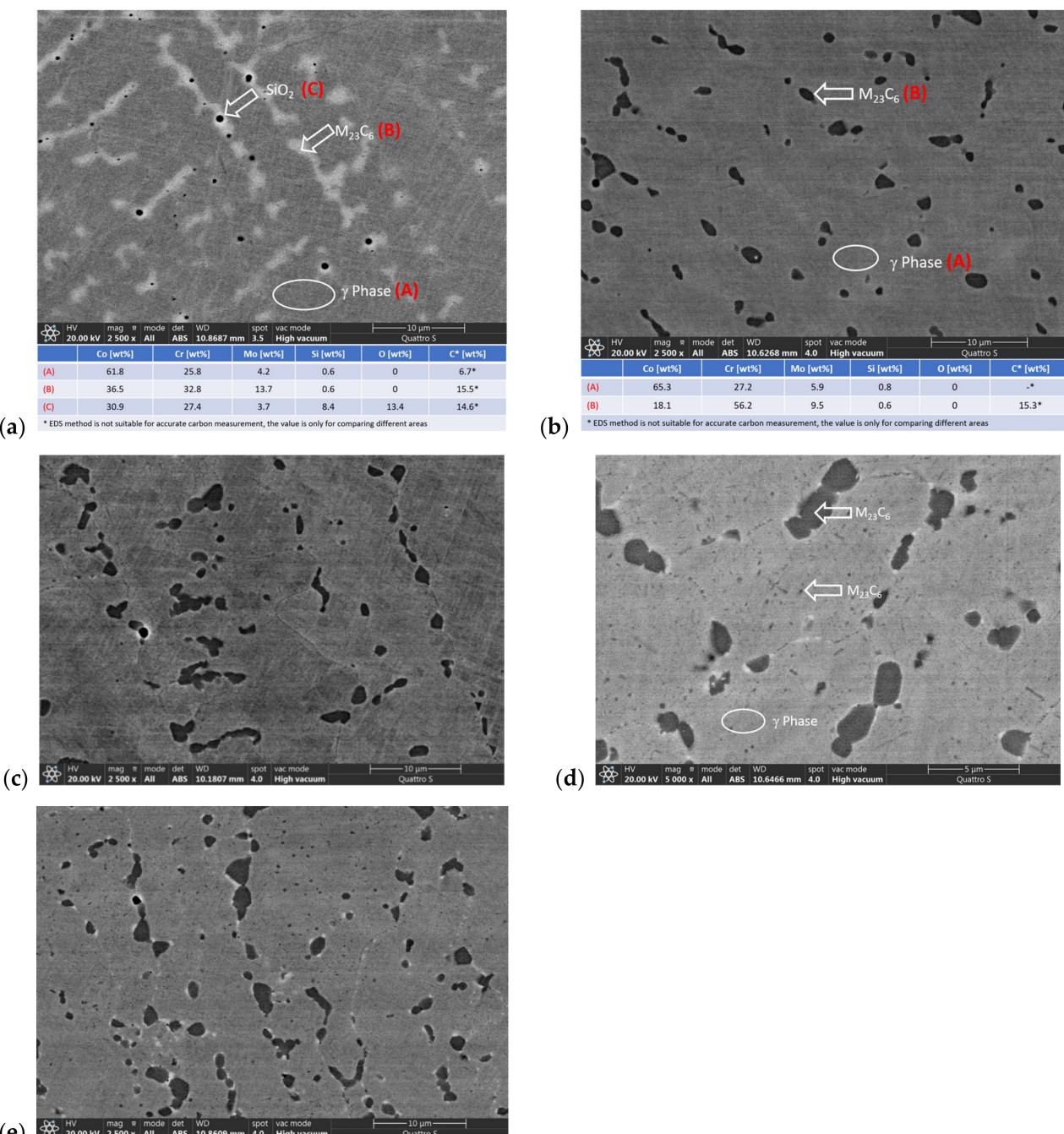

**Figure 2.** Microstructures of the Co-Cr-Mo alloy (**a**) as-cast + EDS, (**b**) quenched + EDS, (**c**) aged at 900 °C for 1 h, (**d**) aged at 900 °C for 3 h and (**e**) aged at 900 °C for 12 h.

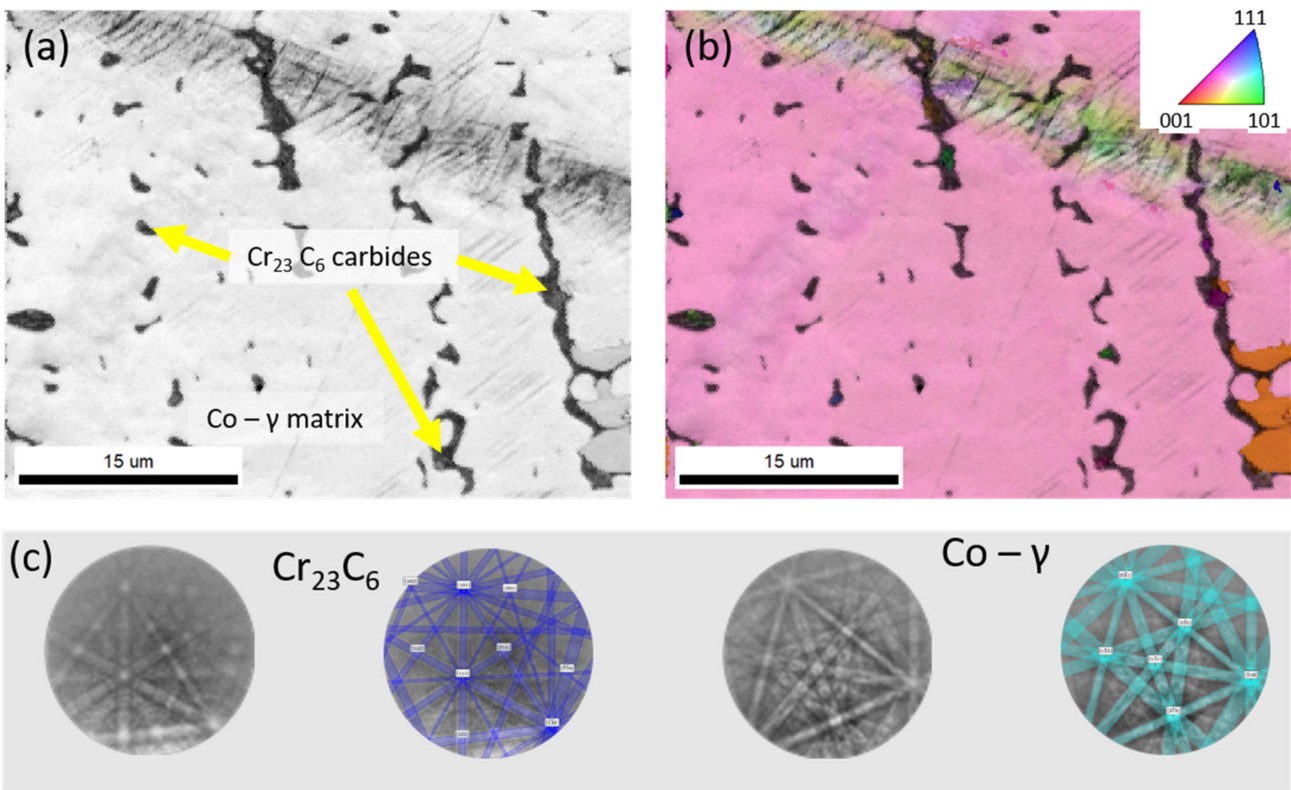

**Figure 3.** EBSD analysis of the Co-Cr-Mo alloy in the as-cast state; (**a**) pattern image quality, (**b**) IPF-Z coloured image and (**c**) representative patterns of the carbide and matrix phase.

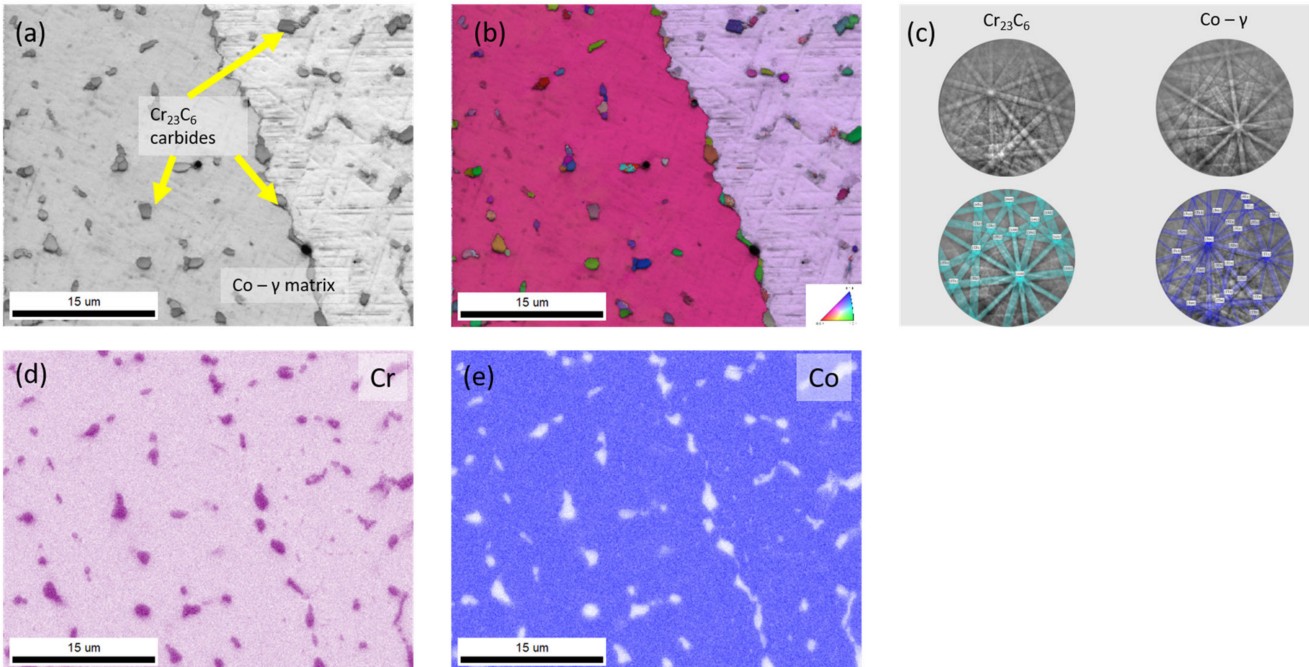

**Figure 4.** EBSD analysis of the Co-Cr-Mo alloy aged at 900 °C for 12 h; (**a**) pattern image quality (**b**) IPF-Z coloured image, (**c**) representative patterns of the carbide and matrix phase, (**d**) EDS signal of Cr and (**e**) EDS signal of Co.

### 3.1.2. Co-Cr-W-Mo Alloy

Compared to the Co-Cr-Mo alloy, this alloy has an addition of W, which together with Mo improves its mechanical and corrosive properties. A chemical analysis also confirmed the presence of Nb, which is usually added because it has a high biocompatibility and a good corrosion resistance. The addition of niobium is also said to cause faster bone growth around an implant [14,16,17]. In the as-cast state, the directional growth of dendrites from the edge to the centre of the specimen was visible (Figure 5a). The γ-matrix phase (face-centred cube—FCC) is a solid solution of cobalt and chromium. The cobalt content was about 60 wt.%, and the chromium content was about 25 wt.%. Carbide particles were found in the interdendritic space and at the boundaries of the crystal grains. Based on the results of the EDS analyses and indications from the literature, the carbides were classified as $M_6C$ and MC (WC). The type of carbides in the microstructure depends on the heat treatment during which the $M_6C$ carbide decomposes into the MC carbide. Since the carbides in this case also contained chromium, molybdenum and niobium, we could conclude that the microstructure contained only $M_6C$ carbides, which are extremely unstable. According to the literature, WC carbides (MC) should only be formed by the decomposition of $M_6C$ if they contain less than 10% of Cr [18]. The mentioned carbides are extremely brittle and have a crystal structure consisting of an asymmetric hexagonal cell [19].

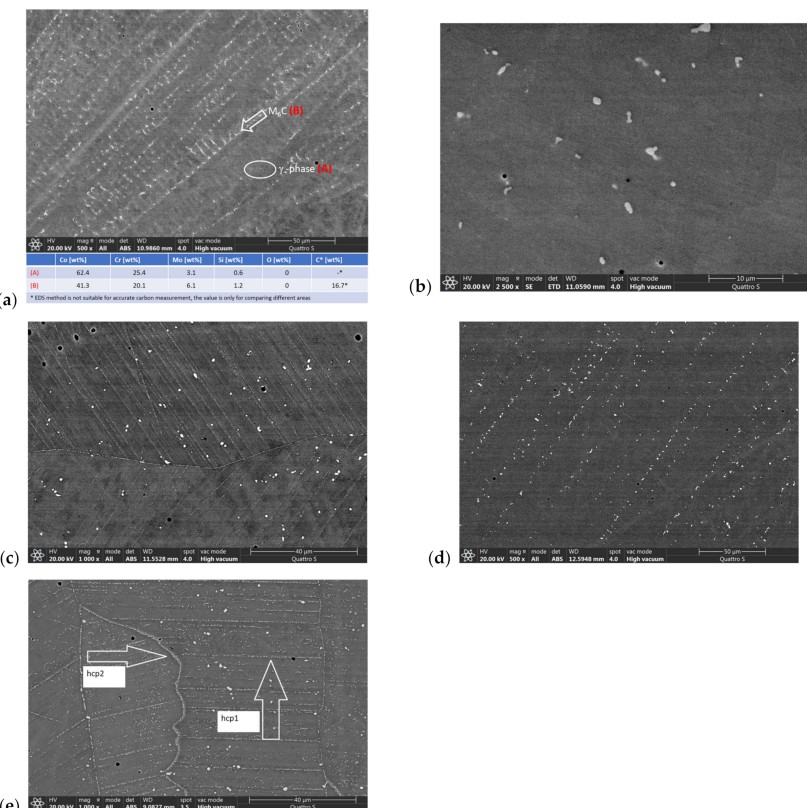

**Figure 5.** Microstructures of the Co-Cr-W-Mo alloy (**a**) as-cast, (**b**) quenched, (**c**) aged at 900 °C for 1 h, (**d**) aged at 900 °C for 3 h and (**e**) aged at 900 °C for 12 h.

After solution annealing and quenching in water, the microstructure was more homogeneous as the dendritic structure was no longer visible, but carbides were still present in the microstructure (Figure 5b). The number of carbides that had not completely dissolved was much lower, and they were also more oval. The EDS analysis of the carbides showed that the content of the element chromium had decreased (from about 20 wt.% to 11% wt.%), while the content of cobalt and especially of tungsten had greatly increased (from 12 wt.% to 23 wt.%). Since the chromium content was above 10 wt.%, it could be concluded that these were still $M_6C$ carbides.

Aging for 1 h had no effect on the chemical composition of the matrix, as it was practically unchanged compared to the quenched state. The EDS analysis of the carbides showed that the content of the element chromium increased (from about 11 wt.% to 16 wt.%), and the cobalt content increased strongly (from 30 wt.% to 41 wt.%). Since the chromium content was above 10 wt.%, it could be concluded from the literature that these were still $M_6C$ carbides. In Figure 5c, it can be seen that during aging a strong crystallographic orientation-dependent precipitation of smaller carbides in straight lines and along the crystal grain boundaries occurred. The literature [4] states that the γ-phase with the fcc crystal lattice transforms to ε-martensite (hcp1) during isothermal annealing at 921 °C. ε-Martensite appears in the form of long straight lines on which carbides precipitate during aging, as shown in Figure 5c.

As can be seen in Figure 5d, a longer aging time (3 h) caused Ostwald ripening of the inclusions formed on the straight lines of the ε-martensite, making them slightly coarser and decreasing their number. The chemical composition and morphology of the undissolved carbides and the matrix did not change significantly.

After 12 h of ageing, we could also observe fine needle-like precipitates formed inside the crystal grains of the γ-phase. The literature [4] states that in addition to ε-martensite (hcp1), the hcp2 phase can also form during aging. This nucleates at the grain boundaries of the γ-phase. Similar to the straight lines of the ε-martensite, carbides could also be seen within the hcp2 phase (Figure 5e). The chemical composition of the matrix remained virtually unchanged compared to that of the quenched state. The EDS analysis of the carbides showed that they contained cobalt and tungsten (the tungsten content was 20 wt.% higher than in the as-cast state). It was also evident from the results that the chromium content dropped below 10 wt.%, from which it could be concluded that WC carbides were formed, which is also consistent with the literature [18]. Figure 6 presents the results of the EBSD analysis. The identified carbides were of the $M_2C$ type, rich in W. The ε phase formed in straight lines, and along these lines, we also observed carbide formation.

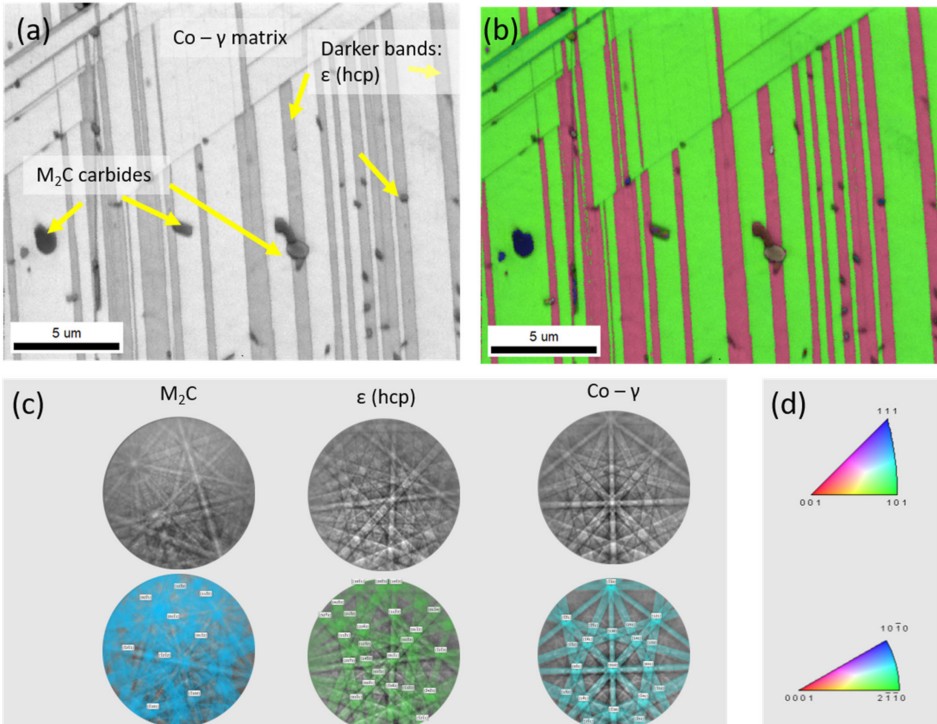

**Figure 6.** EBSD analysis of the Co-Cr-W-Mo alloy aged at 900 °C for 12 h; (**a**) pattern image quality, (**b**) IPF-Z coloured image, (**c**) representative patterns of the carbide, ε, and matrix phase (Co-γ) and (**d**) legend for the IPF-Z colouring.

### 3.2. Hardness

From the results (Figure 7), it was seen that the microstructure of the Co-Cr-Mo alloy in the as-cast state had a higher hardness than that of the quenched alloy. This was attributed to a reduction in the amount of hard carbides that were partially dissolved during solution annealing. The difference in hardness between one- and three-hour-aged alloys was relatively small. This can be attributed to the fact that these two alloys had not yet formed, as many precipitated during the aging process itself, which could affect the strength properties. $M_{23}C_6$ carbides were observed in the one- and-three-hour aged samples. The carbides became rounder in shape with the aging process. They were also precipitated along the crystal grain boundaries, further strengthening the alloy. The alloy that was aged for 12 h achieved the highest hardness. In its microstructure, both $M_{23}C_6$ carbides and smaller inclusions were observed, which had formed during the aging process but could not be analysed by EDS due to their small size. In the microstructure aged for 12 h, brighter carbides, molybdenum-based $M_6C$ carbides, could also be observed at the grain boundaries of the crystal grains. Since the highest hardness was achieved in the alloy that was aged for 12 h, it can be concluded that it was not yet overaged.

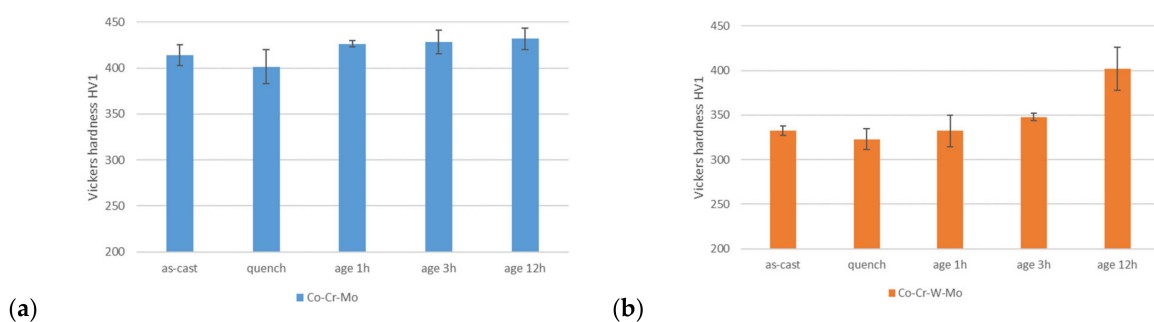

**Figure 7.** Hardness of the investigated samples (HV1): (**a**) Co-Cr-Mo and (**b**) Co-Cr-W-Mo alloys.

As can be seen in Figure 7, the Co-Cr-W-Mo alloy (Figure 7a) had a slightly lower overall hardness than the Co-Cr-Mo alloy. In general, Co-Cr-W-Mo alloys tend to be harder than Co-Cr-Mo alloys due to the presence of hard primary tungsten-based carbides. The effect of precipitation hardening depends on the ageing conditions (temperature and time). From the hardness measurements it can been seen that for the Co-Cr-W-Mo alloy, the peak hardness was not achieved yet.

Similar to the Co-Cr-Mo alloy, the Co-Cr-W-Mo alloy (Figure 7b) also had a higher hardness in the as-cast state than in the quenched state. The reason for this is the same as for the previous alloy: the strength properties decreased after quenching, which was due to the dissolution of hard carbides. In aged specimens, a gradual increase in hardness with the aging time could be observed. In the above alloy, $M_6C$ carbides formed during the ageing process on the basis of W and Mo. The carbides were visibly separated along the straight lines of ε-martensite, which also formed during aging, further strengthening the alloy. The highest hardness was also achieved in this case for the alloy that was aged for 12 h. In its microstructure, small carbide precipitates appeared within the γ-phase, which formed during the aging process and further increased the strength properties.

### 3.3. Wear Properties

The wear properties of the investigated materials are shown in Figure 8, where the specific wear rates of the investigated alloys are also presented. The Co-Cr-Mo alloys showed higher wear in the quenched state than in the as- cast state and exhibited the highest wear (Figure 8a). Aging had a positive effect on wear resistance, as the wear rate decreased with increasing aging time. The sample that was aged for 12 h showed the highest wear resistance. Comparing these results with the hardness measurements, it can be seen that the wear resistance of the alloy increased with its hardness. The SEM analysis

of the wear marks revealed both abrasive and adhesive wear mechanisms. Abrasive pitting was present in all samples, and some cracks were also observed. It was also noted that no carbide particles were extracted from these specimens, indicating that the matrix hardness was high enough to support the load without carbide particle pull out.

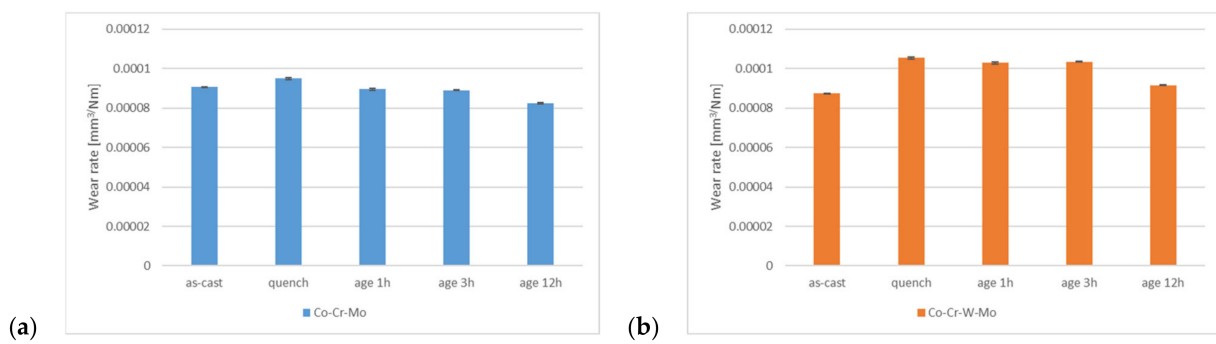

**Figure 8.** Wear rates of the investigated samples; (**a**) Co-Cr-Mo and (**b**) Co-Cr-W-Mo alloy.

For the Co-Cr-W-Mo alloy, on the other hand, the lowest wear rate was measured in the as-cast state (Figure 8b). The highest wear rate was measured for the quenched material, which also had the lowest hardness. The samples that were aged for one and three hours had about the same wear rate (they also had about the same hardness). Figure 9 shows the wear profiles of the as-cast alloy and of the alloy aged for three hours. In both cases, abrasive marks can be seen, but with the difference that larger holes can be seen in the profile of the aged sample (Figure 9d). This indicates that the carbides had fallen out or had been pulled out of the matrix, which means that the matrix was too soft in this case and could not bear the load. The lowest wear rate was found for the sample aged for 12 h. This alloy also had the highest hardness among the Co-Cr-W-Mo alloy samples. In this case, the matrix was hard enough to withstand the loads and prevent the carbide particles from breaking out of it.

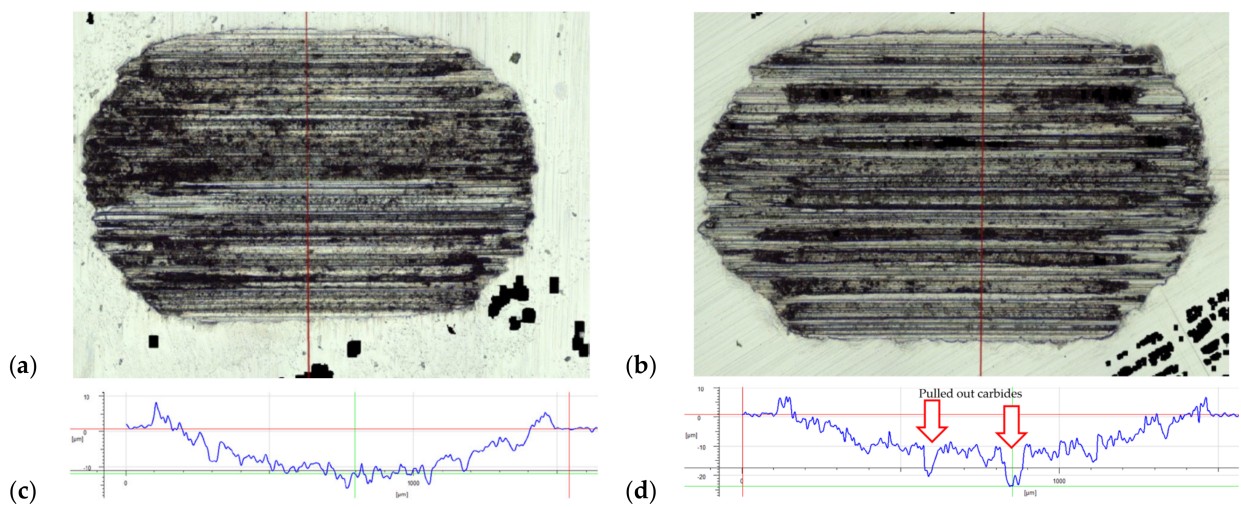

**Figure 9.** Wear tracks and 2D profiles of the (**a**,**c**) as-cast and (**b**,**d**) 3 h aged Co-Cr-W-Mo alloy.

In addition to particle pulling, all samples of the Co-Cr-W-Mo alloy showed signs of abrasive wear, as shown in Figure 10a. Abrasive wear occurs when a harder body or harder particles press against a softer material, resulting in material loss. In our case, three-body abrasion most likely also occurred, meaning that particles formed during the wear process, causing additional wear. Cracks were also observed on the samples. In addition to abrasive wear, traces of adhesive wear were also observed (Figure 10b). Adhesive wear

was confirmed by SEM analysis. The EDS analysis showed that the dark areas were places where aluminium and oxygen were present. From this, we can conclude that the ball used to measure the wear resistance was made of an $Al_2O_3$ compound.

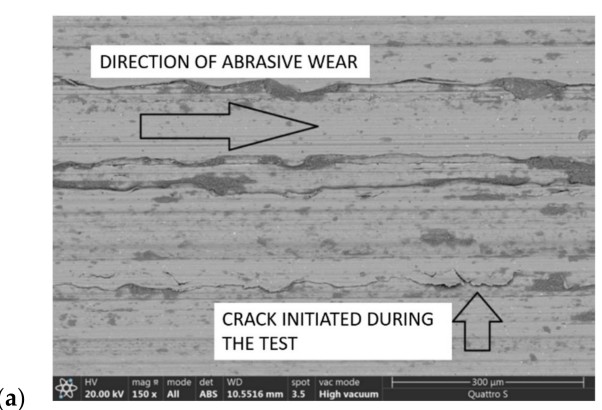 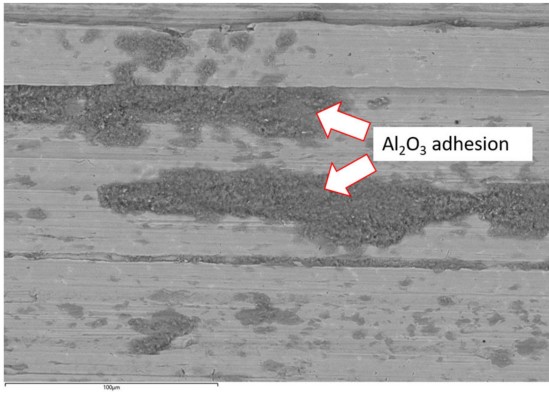

**Figure 10.** SEM analysis of the wear track of the Co-Cr-W-Mo alloy, aged for 12 h showing (**a**) abrasion and cracks, (**b**) adhesion.

Both abrasive and adhesive wear was observed in both alloys. Some cracks were also observed in the wear marks, which were much more visible in the Co-Cr-W-Mo alloy. Although adhesive wear was not as pronounced, it was still observed in both alloys. In both alloys, material was also transferred from the test ball to the material under investigation.

The comparison of the two alloys showed that the Co-Cr-Mo alloy had better wear resistance than the Co-Cr-W-Mo alloy. This can be attributed to the higher hardness of the Co-Cr-Mo alloy. Furthermore, the 2D profiles of the wear traces and the SEM analysis showed the extraction of carbides in the case of the Co-Cr-W-Mo alloy (Figure 11). The higher wear and pull-out indicated that the matrix of the Co-Cr-W-Mo alloy was too soft and could not bear the contact load. In the Co-Cr-W-Mo alloy, the carbides themselves were not capable of preventing wear. Since the matrix was softer, it was not able to hold the hard carbides, and thus the carbides tore from the matrix [20]. This was then reflected in a higher wear.

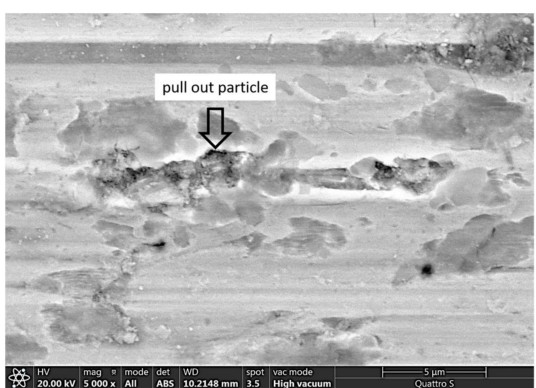

**Figure 11.** SEM analysis of a pull-out particle.

## 4. Conclusions

The influence of heat treatment on the hardness and wear properties of two dental alloys, Co-Cr-Mo and Co-Cr-W-Mo, were investigated. The samples were first solution-annealed (1200 °C/2 h) and quenched and then aged at 900 °C for 1, 3 and 12 h. The resulting properties were also compared with those of the as-cast alloys. Based on the results, the following conclusions can be drawn.

The microstructure of the Co-Cr-Mo alloy in its initial state (as cast) consisted of $\gamma$-phase dendrites and $M_{23}C_6$ carbides rich in Cr and Mo located in the interdendritic regions and at the crystal grain boundaries. During solution annealing at a temperature of 1200 °C, the carbides partially dissolved, while the undissolved carbides acquired a rounder shape. Precipitation of carbides in the microstructure was observed only after 12 h of aging at a temperature of 900 °C. No precipitation of carbides was observed at 1 and 3 h of aging. After 12 h of aging, $M_6C$ molybdenum-based carbides were observed forming along the boundaries of the crystal grains.

The microstructure of the as-cast Co-Cr-W-Mo alloy consisted of $\gamma$-phase dendrites and tungsten-based $M_6C$ carbides. Carbides were also present in the interdendritic regions and at the crystal grain boundaries. The microstructure after solution annealing contained a small amount of $M_6C$ carbides as they were mostly dissolved, and spheroidisation also occurred. After aging at 900 °C, fine carbides based on W and Mo began to separate after only one hour. They precipitated on the straight line of $\varepsilon$-martensite and crystal grain boundaries. After three hours of aging, the carbide precipitates were slightly coarser. After 12 h of aging, the precipitates in straight lines and crystal grain boundaries were even more pronounced. Fine carbide precipitates were also observed within the $\gamma$-phase. A lamellar morphology of the hcp2 phase appeared at the boundaries of the crystal grains, in which WC carbide precipitates could be seen.

The Co-Cr-Mo alloy had an average hardness of 414 HV1 in the as-cast condition; after solution annealing and quenching, it dropped to 348 HV1, and with increasing aging time, it increased to 432 HV1 (12 h). The Co-Cr-W-Mo alloy generally had a slightly lower hardness, i.e., 332 HV1 in the as-cast status, 323 HV1 after solution annealing and quenching, which gradually increased to 393 HV1 (12 h) with an increasing ageing time. The lower hardness of Co-Cr-W-Mo alloy indicated that the peak hardness had not been achieved.

In general, the Co-Cr-Mo alloy exhibited better wear resistance than the Co-Cr-W-Mo alloy, which can also be attributed to the higher hardness of this alloy. Both alloys showed a predominance of abrasive wear and traces of adhesive wear and exhibited the worst wear resistance in the solution-annealed condition. The Co-Cr-Mo alloy exhibited the best wear resistance after 12 h of aging, and the Co-Cr-W-Mo alloy in the as-cast condition. For both alloys, the wear resistance improves with longer aging, followed by an increase in hardness.

In the case of the Co-Cr-W-Mo alloy, a tearing of the carbides from the matrix was observed, resulting in higher wear. This can be attributed to the lower hardness of the matrix, which was not able to carry the contact load.

**Author Contributions:** Conceptualization, M.S. and A.N.; methodology, M.S. and A.N.; validation, B.K., M.Z. and K.Z.; formal analysis, K.Z., M.Z. and B.Š.B.; investigation K.Z., M.S. and B.Š.B.; resources, B.K.; data curation, K.Z.; writing—original draft preparation, M.S. and K.Z.; writing—review and editing, M.S. and A.N.; visualization, M.S., K.Z.; supervision, A.N. All authors have read and agreed to the published version of the manuscript.

**Funding:** This research was partly funded by the Slovenian Research Agency, research core funding No. P2-0050.

**Data Availability Statement:** The data presented in this study are available on request from the corresponding author. The data are not publicly available due to restrictions regarding an on-going funded research.

**Conflicts of Interest:** The authors declare no conflict of interest. The funders had no role in the design of the study; in the collection, analyses, or interpretation of data; in the writing of the manuscript, or in the decision to publish the results.

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
