# Peer review of "Influence of Precipitation Hardening on the Mechanical Properties of Co-Cr-Mo and Co-Cr-W-Mo Dental Alloys"

_metals, doi:10.3390/met13030637_

Round 1

Reviewer 1 Report

The subject matter of the article is interesting, but unfortunately the article still needs a lot of work. Especially the Introduction needs a lot of correction. My criticisms are as follows:

The introduction is badly written. Most of the text is off-topic. Statements that Co-Cr-Mo alloys are of great importance for medical applications and that implants made of them are subject to destruction should be shortened to 3-4 sentences. Information about the structure of these alloys is important.

The description of the structure of alloys and precipitates is chaotic and imprecise. This part definitely needs improvement. I suggest that the authors prepare a plan for this part, and then, according to this plan, prepare a description (one paragraph to one point of the plan). In its current form, information is provided chaotically. Please provide such a plan in response to the review.

Line 78 (and Line 231). Please explain what is the difference between the ε-phase and hexagonal close-packed (HCP)? Do not use double naming, because it makes it difficult for the reader of the article to understand the text.

Please use g-phase and e-phase or HCP and FCC consistently. Can also be g(fcc) and e(hcp). But always the same. They cannot be used interchangeably.

Line 160 Do not provide information about results that are not presented in the article (results of EDS analyses). Please complete the spectra or remove from the text.

Line 191. The signature is missing whether it is SE or BSE. Why are the carbides light in Fig. 2a and dark in the other pictures?

Please complete the results with XRD tests. This is a common and cheap technique, and without it, inferring the presence of carbides is questionable. Why do research if phase identification is based only on previous literature data?

Please explain the meaning of the abbreviation PCK (Fig. 2.)

Please improve the presentation of EBSD results. The drawings should represent (as a minimum): image quality (IQ) maps, inverse pole figure (IPF) maps and phase maps. You can show example EBSD Kikuchi patterns, but necessarily with the solution (drawn Kikuchi lines). A legend should be added to the drawings (the meaning of the color in a given drawing). All drawings should be described.

Line 187. Why was no EDS analysis done for the sample aged at 900 °C for 12h at higher magnification? There would then be no need to guess whether the small carbides contain Mo.

Line 230 The authors wrote that "It seems that there was a strong crystallographically dependent precipitation of smaller carbides in straight lines and along the crystal grain boundaries." Referring to [15], which is probably a mistake. Please provide the correct reference.

I would like the authors to explain why the long straight lines of e-phase are present in Co-Cr-W-Mo alloy samples aged at 900 °C, but not in Co-Cr-Mo alloy samples. How should this affect the mechanical properties? It would also be valuable to determine the percentage of the e and g phases in the tested samples (based on XRD). This will allow you to better justify your wear properties considerations. The hardness of the two phases is different.

Line 281 Please complete the phase symbol.

Fig. 10b. The figure shows "Mesto 1", "Mesto 2". This is not described in the text. Please correct.

Information about the manufacturer of the alloy should appear at the beginning of the article, not just in Conclusions.

The conclusions are not convincing. The authors suggest that the matrix of Co-Cr-Mo alloy contains only (mainly) the gamma phase, which is ductile. In the case of the Co-Cr-W-Mo alloy, the share of the harder epsilon phase is higher. At the same time, the authors write that higher wear in that case can be attributed to the lower hardness of the matrix.

Author Response

We would like to thank the reviewer for its thoughtful and insightful comments. We believe that with your help we were able to improve the article. We have followed the comments from the reviewers and made some revisions accordingly, as specified below. (The answer is written in upright letters, and the newly written text in italics In the manuscript revised parts are marked with red color). 

The answers are in the attached file

Reviewer 2 Report

Dear authors, 

The aim of this manuscript is interesting but some details must be improved. It is not clear if the tested materials are commercial or made specially for this study.

- Are the compositions presented in Table 1 in agreement with the nominal compositions of these alloys?

- Why did you chose this temperature x times to perform these heat treatments?

- Some Figures are not cited in the text. Please revise.

-Why solution annealing samples still show M23C6? Have you tried other temperatures x time?

Author Response

(The authors gave the same response as above.)

Round 2

Reviewer 1 Report

I have no comments